# CoGTEx: Unscaled system-level coexpression estimation from GTEx data forecast novel functional gene partners

**Miguel-Angel Cortes-Guzman[1], Víctor Treviño[1,2]***

1 Tecnologico de Monterrey, Escuela de Medicina, Bioinformática, Monterrey, Nuevo León, México,
2 Tecnologico de Monterrey, OriGen Project, Monterrey, Nuevo León, México

* vtrevino@tec.mx

## Abstract

### Motivation

Coexpression estimations are helpful for analysis of pathways, cofactors, regulators, targets, and human health and disease. Ideally, coexpression estimations should consider as many diverse cell types as possible and consider that available data is not uniform across tissues. Importantly, the coexpression estimations accessible today are performed on a "tissue level", which is based on cell type standardized formulations. Little or no attention is paid to overall gene expression levels. The tissue-level estimation assumes that variance expression levels are more important than mean expression levels. Here, we challenge this assumption by estimating a coexpression calculation at the "system level", which is estimated without standardization by tissue, and show that it provides valuable information. We made available a resource to view, download, and analyze both, tissue- and system-level coexpression estimations from GTEx human data.

### Methods

GTEx v8 expression data was globally normalized, batch-processed, and filtered. Then, PCA, clustering, and tSNE stringent procedures were applied to generate 42 distinct and curated tissue clusters. Coexpression was estimated from these 42 tissue clusters computing the correlation of 33,445 genes by sampling 70 samples per tissue cluster to avoid tissue overrepresentation. This process was repeated 20 times, extracting the minimum value provided as a robust estimation. Three metrics were calculated (*Pearson*, *Spearman*, and *G-statistic*) in two data processing modes, at the system-level (TPM scale) and tissue levels (z-score scale).

### Results

We first validate our tissue-level estimations compared with other databases. Then, by specific analyses in several examples and literature validations of predictions, we show that system-level coexpression estimation differs from tissue-level estimations and that both contain valuable information reflected in biological pathways. We also show that coexpression

**Data Availability Statement:** All relevant data are within the manuscript and its Supporting Information files.

**Funding:** The author(s) received no specific funding for this work.

**Competing interests:** The authors have declared that no competing interests exist.

estimations are associated to transcriptional regulation. Finally, we present CoGTEx, a valuable resource for viewing and analyzing coexpressed genes in human adult tissues from GTEx v8 data. We introduce our web resource to list, view and explore the coexpressed genes from GTEx data.

## Conclusion

We conclude that system-level coexpression is a novel and interesting coexpression metric capable of generating plausible predictions and biological hypotheses; and that CoGTEx is a valuable resource to view, compare, and download system- and tissue- level coexpression estimations from GTEx data.

## Availability

The web resource is available at http://bioinformatics.mx/cogtex.

## Introduction

All cells in an organism must express several genes coordinately to execute particular and complex biological functions [1]. This concept is known as coexpression. Coexpression can be used to assembly biological pathways or modules [2, 3], to focus research on the discovery of unstudied but highly coexpressed genes [4], to estimate possible regulators [5, 6], to evaluate genes targets of regulation by a transcription factor [7], to build and use biological networks [8], to compare altered networks in diseases [9, 10], and to compare coexpression differences between species [11].

Coexpression is typically estimated from many independent samples of the same cellular origin, assuming that the cell mixture is homogenous or from time-series analyses [12]. However, when different cellular origins are used (e.g. different tissues), the estimation can be either calculated per specific origin or integrated into an overall measure as a weighted coexpression [13, 14]. We will focus on coexpression calculations considering multiple tissues.

For coexpression, organism-specific databases and tools exist [15–21]. For humans, there are tissue-specific databases [22, 23] and general databases that consider many tissues or cell types [13, 14]. As tools and databases, larger compendiums of tissues or cell types are more attractive to a broader scientific and medical community. GTEx is a recent resource where samples from over 800 post-mortem donors in about 50 tissues were systematically sequenced and made available [24]. Despite some efforts [25–29], researchers have no resources to view and analyze the data of GTEx gene coexpression estimations easily.

Although databases and tools for human coexpression have been helpful [14, 20], there are technical details that have limited the coexpression analyses, particularly in the case when many tissues or cellular origins are used. Historically, the most common view of coexpression when many tissues are used involves normalizing tissues or weighting schemes to compensate for tissue over-representation [13]. This type of coexpression is implemented by transforming the data into standardized scores per tissue or cell type, such as z-scores, and then estimating coexpression by linear correlation or principal components of the aggregated data from tissues or cell types [20, 28, 30]. One of the underlying assumptions is that the direction of coexpression (slope, for instance) is similar in all tissues, and intrinsically this standardized local slope is the most important component of the overall coexpression. This view is represented in

panels *v* and *vi* of Fig 1 where the scale of the original data has been standardized into z-scores (tissue level coexpression). However, coexpression can be observed theoretically in broader scenarios. For example, when gene expression levels are clearly different across tissues (panels *i* to *iv*), but they are barely correlated within the tissue (panel *iii*), are not consistently correlated among tissues (panel *iv*), or are not linearly correlated (panel *iv*). This type of non-standardized coexpression may suggest another substantial component of inter-tissue gene expression control in addition to the intra-tissue control, which is commonly revealed by standardized scores. It could be key to advancing our understanding of gene expression regulation in a broader, organism-wide sense. This type of coexpression will be referred to here as "system-level" because the expression level is not scaled and is compared along all available cellular origins of the organism (other designations could be "organism-level", "genome-level", or "inter-tissue level").

Contrary to the standardized tissue level coexpression, the coexpression at the system level, to our knowledge, has not been studied systematically. This paper proposes a robust procedure to estimate both system- and tissue-level coexpression from GTEx data, show some comparisons, and provide a web tool and database, CoGTEx, to share and explore the results. To provide comparable and robust results over both scenarios, we first filtered the GTEx samples generating clear, unambiguous tissue clusters, then sub-sampled all tissue clusters (n = 20 times) at the same number of samples per cluster (n = 70) to calculate three metrics of coexpression at the system- and tissue levels. We show that our calculations at the tissue level are similar to the estimations available in other databases. More importantly, we show that coexpression gene pairs at the system level differ from those estimated at the tissue level and that the number of coexpressed gene pairs at the system level is also abundant. We also show that a small fraction of the system-level coexpression may show a non-linear behavior. We discuss that some artifacts may be present at the system-level estimations for tissue-specific genes. Our view and results may open new opportunities for research in the biology of health and disease.

## Results

A summary of the CoGTEx workflow is depicted in Fig 2. From the v8 data in the GTEx portal (https://gtexportal.org/) consisting of 56,200 genes and 17,382 samples from 54 tissues, we performed initial cleaning, filtering, global normalization, and batch correction resulting in 33,445 genes and 16,651 samples from 47 tissues (see Methods). Then, we purified samples within tissues removing those mixed among "similar" tissues ensuring an unambiguous representation of distinct human gene expression tissue states, which generates 42 tissue clusters. The cleaned data was used to robustly estimate system-level inter-tissue correlations extracting the minimum correlation (or statistic) among 20 rounds of 70-random-samples-per-cluster. We calculated three coexpression estimations (*Pearson*, *Spearman*, *G-statistic)* [31, 32] in two modalities, at absolute absolute-level gene expression (system level) and relative-level gene expression z-scores (tissue level). Additionally, we estimated the *Pearson* correlation in 3 GTEx commonly used covariates (Sex, Age, and Ischemia). See the Methods section for details. The following sections will describe the results and validations of each process and provide examples of their utility.

### Enrichment of biological terms supports coexpression estimations

Since one of the goals of coexpression analyses is to find possible functional partners. It is common to evaluate coexpressed genes with gene-pairs from known biological networks [28, 33]. Therefore, to support the estimations of the provided associations, we reasoned that top coexpressed genes must be enriched by known biological pathways [28]. Thus, for any gene, we

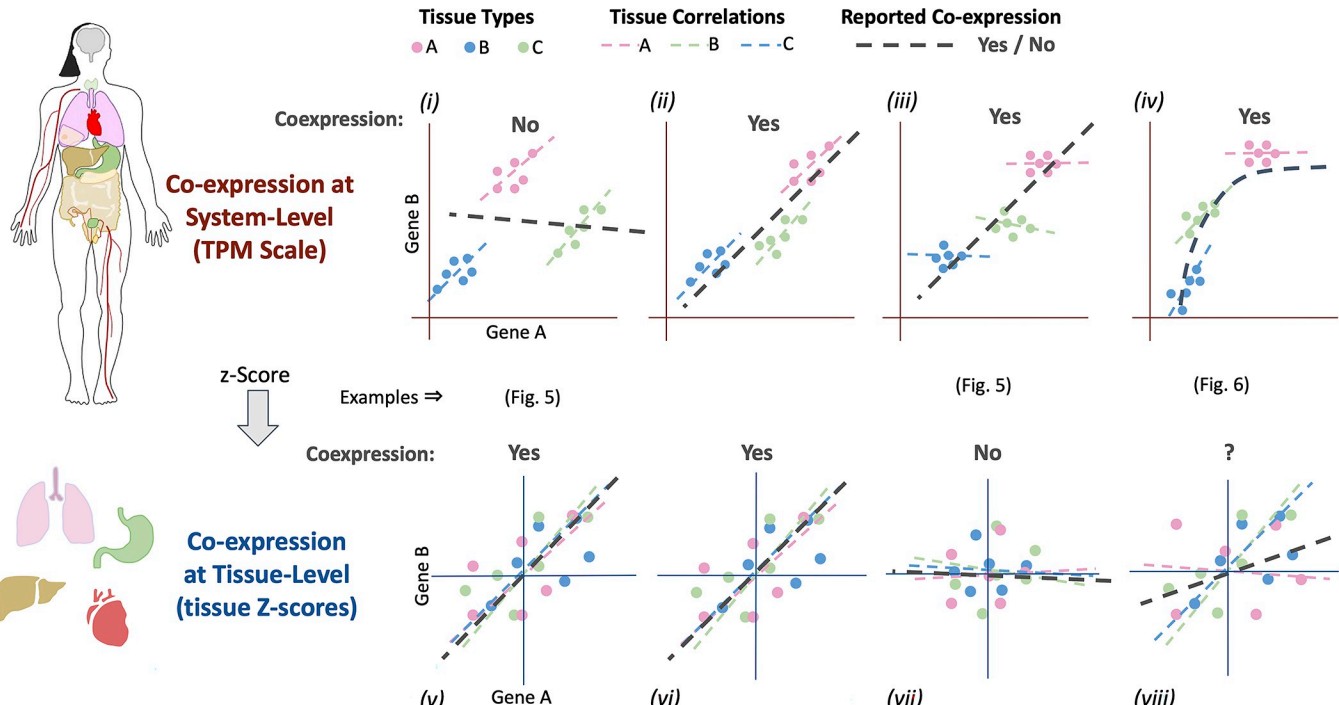

**Fig 1. Coexpression scenarios.** Coexpression has been calculated and is well known from intra-tissue-centered data (bottom scenarios from *v* to *viii*, generally in z-score alike procedures). Nevertheless, coexpression at their whole expression range across tissues or inter-tissue coexpression is not commonly analyzed (in Transcripts Per Millon scale, for example, top scenarios from *i* to *iv*). This estimation corresponds to a whole-organism or system-level view of coexpression, opening new opportunities for research and comprehension of gene function and regulation. Examples of discrepancies among system vs. tissue level estimations are indicated in further figures.

calculated the pathway associations between its top 5% most coexpressed genes (1672 genes, absolute coexpression) compared to the bottom 5% less coexpressed gene (those genes closer to zero coexpression). We used ~200 canonical KEGG pathways [34] assessed by a hypergeometric test counting those pathways whose p-value $< 0.05$ (the p-value is used as a surrogate indicator of biological content). We used the minimum coexpression matrix of each metric and modality. The results are shown in Fig 3A. For example, for *Pearson* TPM, the median is 7 associated KEGG pathways from the top 5% coexpressed genes. Moreover, 25% of the genes show more than 13 pathways associated. In comparison, using 5% of bottom coexpressed genes, basically those showing no correlation, the median is only 3 pathways and marginal number of genes show more than 10 pathways associated. Therefore, overall, Fig 3A reveals that the distributions of the number of associations of top genes (red) are clearly more associated with KEGG canonical pathways than the bottom no coexpressed genes (blue) in the six comparisons.

To further assess the information provided by coexpressed genes, we used the GIANT database [35]. GIANT contains a collection of tissue-specific gene networks where an estimated probability of association links each gene pair. Therefore, we used ~145 module networks and their estimated gene pair probability to determine positive ($p > 0.995$) and negative ($p < 0.005$) GIANT association and assessed whether the gene coexpression correlation across the whole range of values from our estimations could explain these GIANT associations. The results of the distribution of the area under the ROC (AUROC) per gene are shown in Fig 3B. Clearly, most of the genes-pairs show non-random AUROC, suggesting a correlation between our coexpression estimations and GIANT tissue-specific networks.

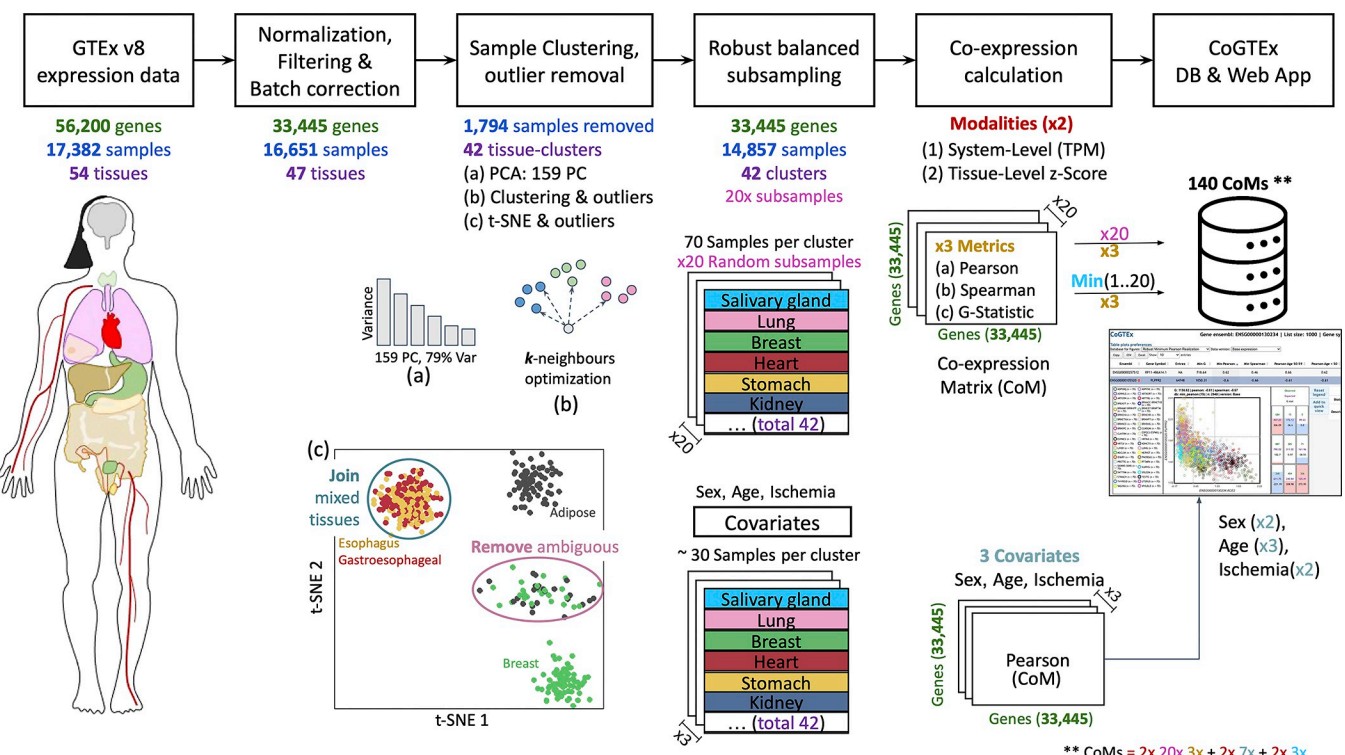

**Fig 2. CoGTEx data processing workflow.** GTEx samples were clustered by similitude and cleaned by removing samples showing some degree of tissue mixture, contamination, or confusion. The 42 tissue clusters were used to robustly estimate the minimum correlation from 20 random 70-sample subset selections per cluster. Pearson, Spearman, and G-statistics are calculated as coexpression measures in two modalities, at system-level gene expression and tissue z-scores. Additionally, the correlation among Sex, Age, and Ischemia co-variates are also computed. The databases are available via a web application at http://victortrevino.bioinformatics.mx:8080/cogtex. The 140 correlation matrices estimated result from 2 modalities, 20 random subsamples, 3 metrics, plus 2 modalities in 7 levels of covariates and the 3 minimums of 2 modalities. See Methods for more details.

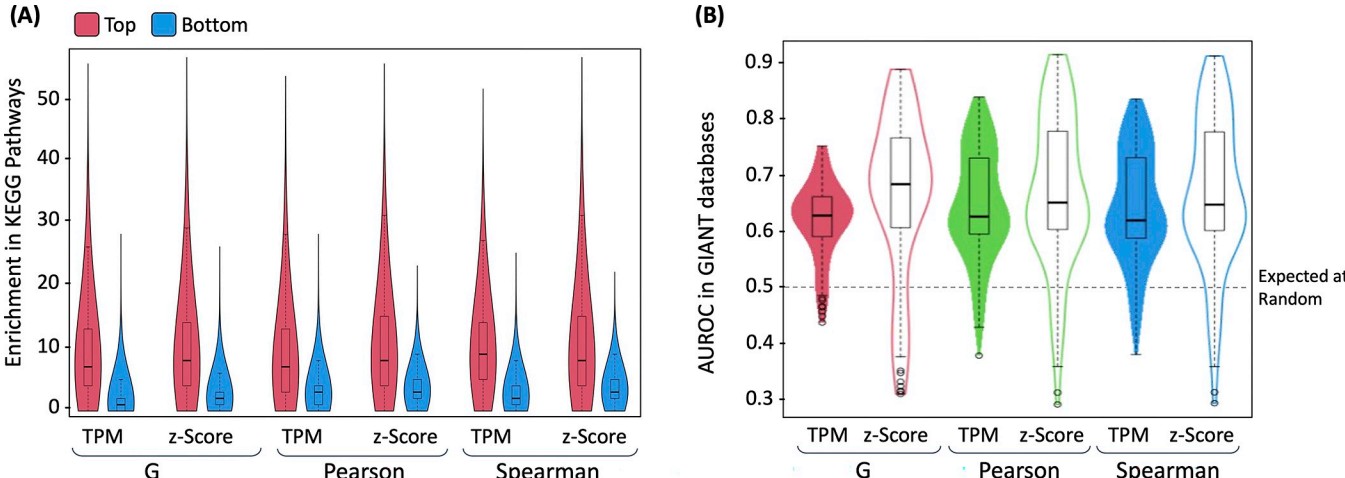

**Fig 3. Pathway and network enrichment analysis.** Panel A shows the number of associations (hypergeometric test $p < 0.05$) to KEGG pathways of the top 5% of most coexpressed genes per every gene (absolute correlation) compared to the bottom 5% (which should represent zero correlation). Each boxplot is composed of the number of associations to ~200 KEGG pathways of the 33,445 genes considered. Panel B shows the AUROC per gene classifying positive and negative GIANT gene pairs from tissue-specific networks.

In both analyses shown in Fig 3, we noted slightly higher indicators for the z-score modality than for the system-level TPM modality.

## Comparisons with other coexpression estimations

Some databases have been proposed for coexpression analyses in humans, mainly from microarray technologies [14, 20, 22, 23, 36, 37]. Some are focused on specific tissues and are not suitable for comparisons at the system level [22, 23, 37]. We used COXPRESdb because it contains almost every tissue sampled by microarrays, has been updated for more than ten years, and recently included RNA-Seq data [20, 38]. COXPRESdb performs a coexpression calculation based on Principal Components and zero-centering, which is similar to our estimation of the z-score modality. To compare the information provided by our estimations with that of COXPRESdb, we performed two experiments using the top 100 ranked coexpressed genes per gene from our estimations. We used only those genes in COXPRESdb, corresponding to 17,871 genes. In the first comparison, we estimated the number of genes overlapping with the top 100 from COXPRESdb along the 17,871 genes. The results shown in Fig 4A demonstrate that our estimations based on the z-score modality are more similar to those from COXPRESdb, with an approximate median of 15 genes overlapping among the top 100 irrespective of the metric. More similarity in the z-score is expected because of the analogous procedure performed by COXPRESdb. Overall, the overlap for most genes is significant, suggesting that estimations are congruent in the z-score. The discrepancies should therefore correspond to differences in input databases and the specific procedures for computations. We noted lower overlap (median of ~7 genes) for the system-level TPM modality, suggesting larger discrepancies in

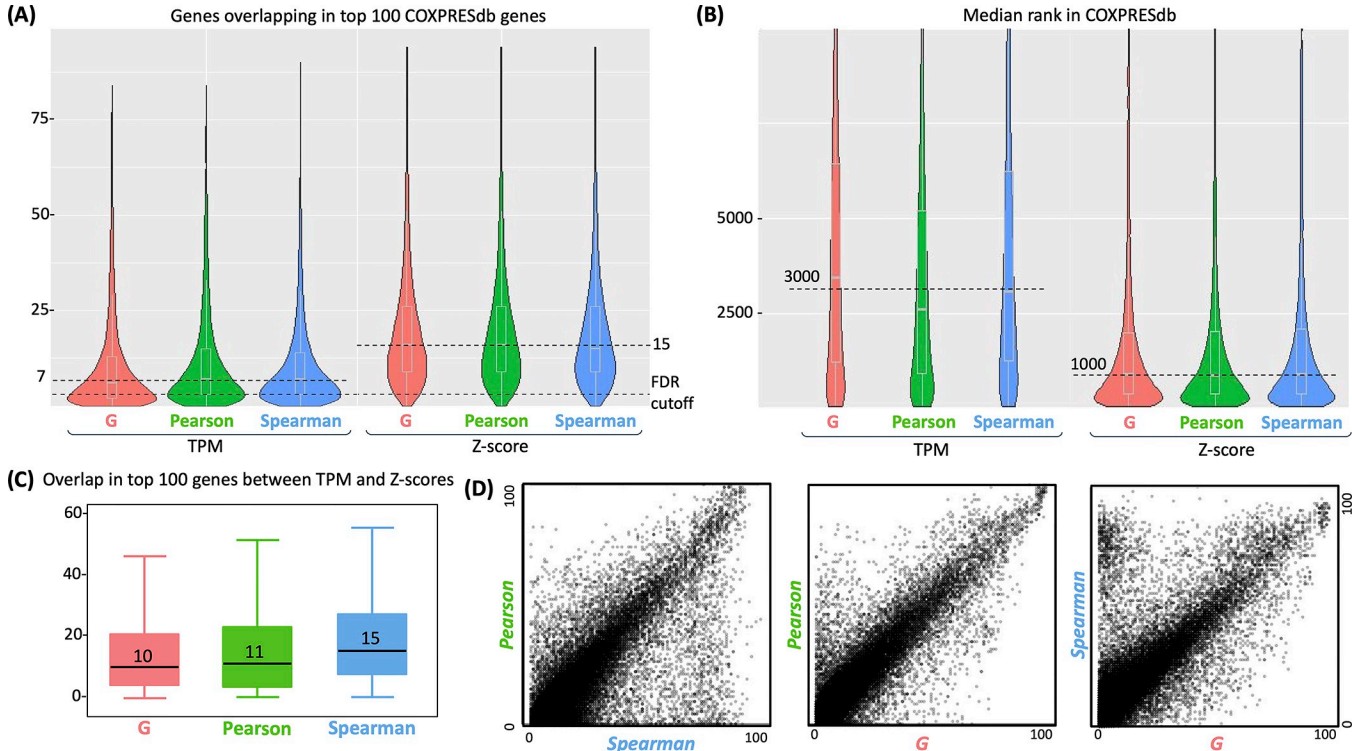

**Fig 4. Comparisons of coexpression estimations.** Panel A shows the distribution of the number of overlapped genes among the top 100 with COXPRESdb. Panel B shows the distribution of the median rank for the top 100 genes in COXPRESdb. Panel C shows the overlap within CoGTEx between system and tissue levels (TPM and Z-scores, respectively) of the top 100 genes. Panel D shows the similitude between overlaps in Panel C. Each dot represents a gene.

the top 100 genes compared to COXPRESdb. Because this comparison was limited to the top 100 genes in COXPRESdb and, in our view, the TPM modality is essential to grasp system-level regulation, we performed an extended comparison estimating the median rank given by COXPRESdb of the top 100 genes from our estimations. If both estimates were equal, the expected median rank would be 50. The results show that the median rank is around 1,000 and 3,000 for the z-score and TPM modalities, respectively (Fig 4B). Analyses for 500 and 1000 top number of genes are provided in S6 Fig in S1 File. These results suggest that the TPM is only slightly similar to COXPRESdb ranks. Consequently, most top genes from TPM will be hidden in the rank provided by COXPRESdb and presumably in all those databases whose estimations are based on tissue-level coexpression.

## System- and tissue-level estimations provide different information

To show a comparison between system- and tissue-level coexpressions, we used the ranks of coexpressed genes for TP53, a highly studied gene (Fig 5). The gene PAICS is the 3rd most coexpressed gene at the tissue level but not coexpressed at the system level (rank 20181). This comparison corresponds to scenarios *i* vs. *v* in Fig 1, representing a well-known scenario based on tissue-level estimations. Nevertheless, the TP53 coexpressed genes SH3GL2 and LTBR (shown in Fig 5 and ranked 2[nd] and 4[th], respectively) are clear examples of coexpression only observed at the system level. Indeed, all the top 10 ranked genes at the system level behave like

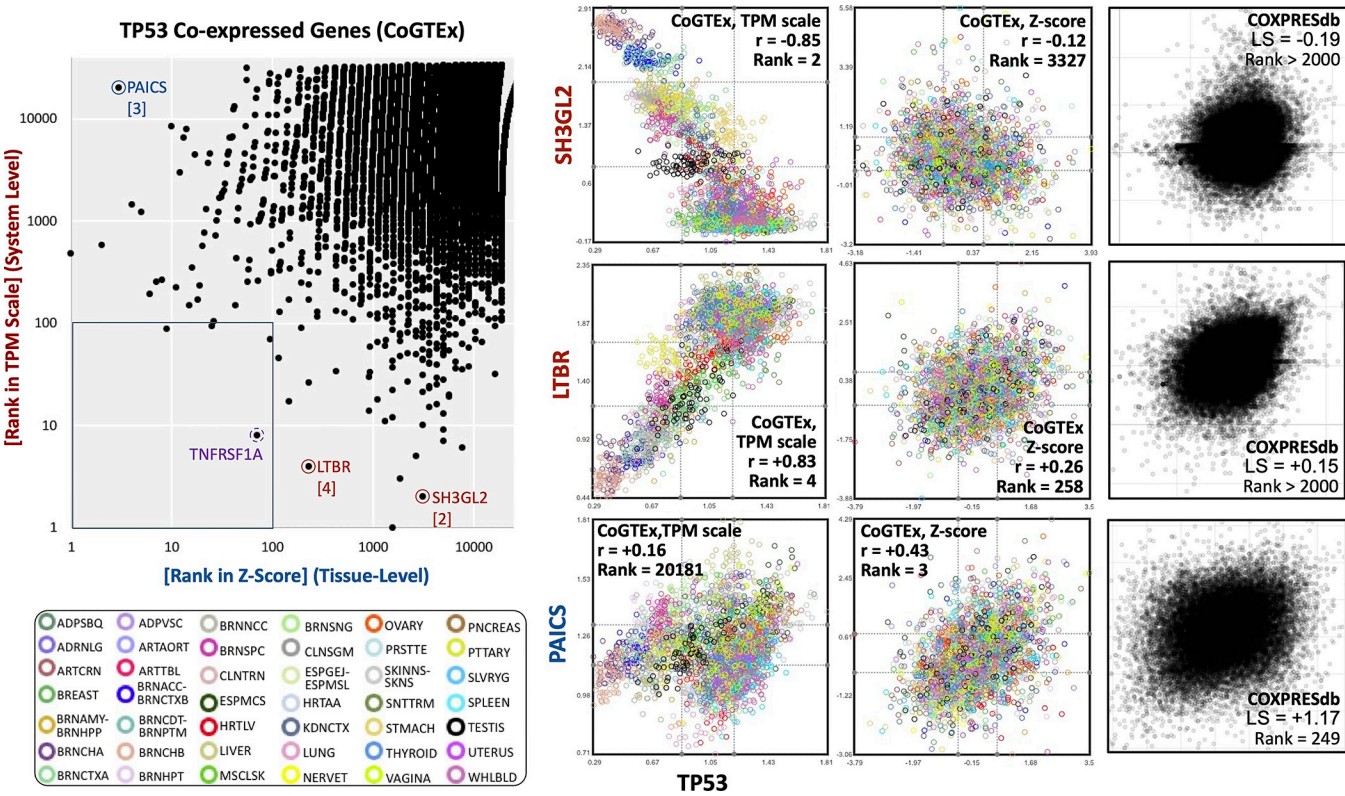

**Fig 5. Comparison of system- and tissue- level coexpression for the TP53 gene.** Left scatter plot shows the coexpression ranks of all genes highlighting specific examples shown at the right. Note the low overlap in ranks among the top 100 (within the square). TNFRSF1A gene (marked with dotted circle is shown in S1 File). At the right, the coexpression estimations for the 3 genes marked are shown. Three estimations of coexpression are shown in columns for comparisons corresponding to system- and tissue level in our results, and the equivalent to tissue-level from COXPRESdb. Point colors correspond to different tissues as indicated (see S1 File for details).

the SH3GL2 and LTBR, as the scenarios *iii* vs. *vii* in Fig 1, where a gene pair is highly coexpressed at the system level, but low coexpressed at the tissue level. Moreover, among the top 100 genes for TP53, only 4 genes are shared between system- and tissue-level coexpression, representing scenarios *ii* vs. *vi* in Fig 1. An example of similar coexpression is TNFRSF1A, ranked 8th at the system level and 75th at the tissue level (S1 Fig in S1 File). Compared to COXPRESdb, our tissue level estimation for TP53 shares 18 genes among the top 100. This result suggests that gene overlaps estimated at the same level (z-scores) between databases are far superior to comparisons at different levels within the same database (which used the same pipeline and input data). We compared the number of shared genes among system- and tissue-level coexpressions for all genes in our estimations for confirmation. The results show that the median shared genes among the top 100 are 10, 11, or 15 for *G-statistic*, *Pearson*, and *Spearman* metrics, respectively (Fig 4C and S5 Fig in S1 File). These results suggest that system-level and tissue-level coexpressions are highly dissimilar, and the difference cannot be attributed to different pipelines or input databases.

## Non-linear estimations at the system level may reveal exceptional tissue expression

Because current databases do not estimate coexpression at the system level, possible non-linear associations have been hidden from researchers. We noted that some system-level associations could not be efficiently explained by linear models. Thus, to detect non-linear associations, instead of imposing a continuous non-linear model such as polynomials or any other, we opted to use a *G*-statistic contingency table inspired by the Maximal Information Coefficient [39] (MIC). We fitted three expression levels (low, middle, and high) to every gene and then estimated the coexpression as those *G-statistics* higher than expected (see Methods). We initially noted that the vast majority of associations were similar to those provided by *Pearson* and *Spearman* but also that some deviations exist. To show an example, we used the top 1000 highest *G-statistic* for the BRCA2 gene, as shown in Fig 6A. It is clear that, as expected, the higher the *G-statistic*, the higher the absolute *Pearson* estimation suggesting that most of the associations are easily explained by a linear model. Nevertheless, some associations appear higher in the *G-statistic* than the linear tendency obtained by *Pearson*. The four examples in Fig 6 clearly show that these high-*G*-low-*Pearson* associations move away from linearity. The combination of BRCA2 and SPTBN4 expression looks like mutual exclusive (or XOR operation), where only one is expressed across tissues rather than the common concept of coexpression. Similarly, BRCA2 and GPR19 appear mutually exclusive except for a couple of tissues. The association of BRCA2 with CASP4 and MFF seems clearly non-linear for most tissue clusters except for testis (black dots). Indeed, many of the discordant values between the *G-statistic* and the *Pearson* seem to be caused by exceptional tissue clusters that move away from the overall tendency, which increases the *G-statistic*, such as in CASP4, MFF, and GRP19. These tissue exceptions are also interesting because they may reveal more specific modes of regulation or function. We added a tool to analyze the ranks of computed metrics to identify these exceptions (option in Analyze tab within the gene page).

## Estimated system-level coexpressions can be validated

Our system-level coexpression implicitly provides a hypothesis generation platform. Thus, we focused on functional proofs that agrees with coexpression in the literature. We used two approaches. First, we looked at the top coexpressed genes for BRCA1 and TP53 to show that top coexpressed genes barely studied can be biologically important. Second, we used recent publications in PubMed to show that current findings agree with results from our estimations.

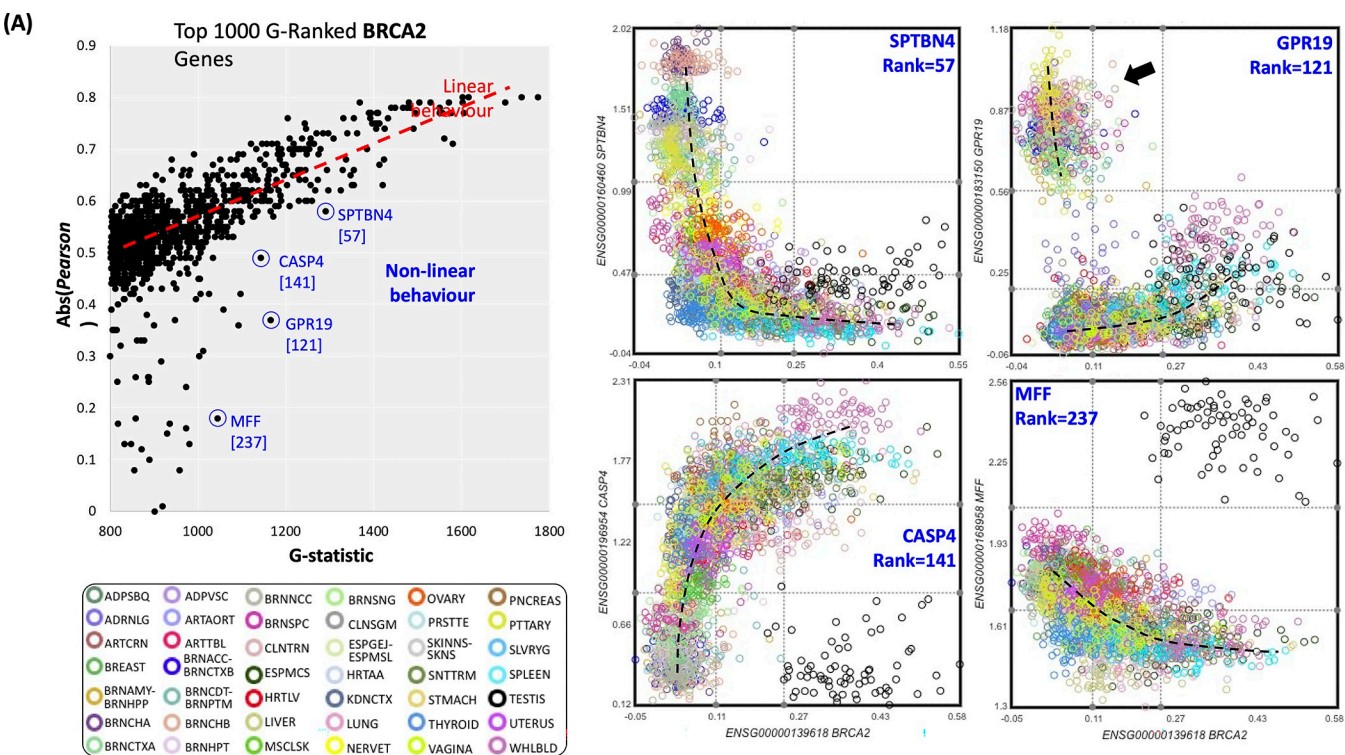

**Fig 6. Examples of non-linear or exceptional cases.** Panel A compares the *G-statistic* and absolute Pearson estimations for the top 1000 coexpressed genes for BRCA. Marked genes show examples of increased *G-statistic* generated by exceptional tissue clusters. Point colors correspond to different tissues as indicated (see S1 File for details).

The PubMed query was run on September 16th, 2022 (top 1000 publications using the query "(gene[TIAB] OR genes[TIAB]) AND expression[TIAB] NOT review") and concentrated on titles including one or two genes, suggesting functional biological assays, and in humans. The following paragraphs and Table 1 show examples of this analysis.

**BRCA1-ASPM.** Recently, it was discovered that ASPM binds BRCA1 and prevents ubiquitination and following degradation by HERC2 [40]. We observed a high system-level coexpression between ASPM and BRCA1 (rank 78), which is evident in most tissues (S2A Fig in S1 File). However, the expression of BRCA1 and ASPM in the thyroid and heart, among other

**Table 1. Examples of coexpressions from literature.**

| Genes | System Level Cor / Rank | | Tissue Level Cor / Rank | | Co-citations | PMID / Year / Summary |
|---|---|---|---|---|---|---|
| BRCA1-ASPM | 0.46 | 78 | 0.27 | 185 | 5 | 34142045 / 2021 / ASPM stabilizes BRCA1 function preventing its degradation |
| BRCA1-HMMR | 0.55 | 1 | 0.31 | 61 | 9 | 22110403 / 2011 / BRCA1-HMMR interplay regulates the epithelial apicobasal polarity |
| TP53-CLIC1 | 0.82 | 10 | 0.28 | 146 | 2 | 32905514 / 2020 / TP53 activates CLIC1 expression |
| DDX58-STAT3 | 0.65 | 55 | 0.14 | 555 | 10 | 36004488 / 2022 / DDX58 suppress cell proliferation by activating STAT3 in colon cancer |
| TWIST1-PDGFRB | 0.64 | 37 | 0.26 | 51 | 4 | 33845139 / 2022 / TWIST1 binds PDGFRB promoter to activate transcription and increasing cell proliferation and migration |
| ERG-(Top1000) | 0.81-0.55 | 3-941 | 0.66-0.08 | 1-864 | - | 34229994 / 2021 / 22 of 30 up-regulated genes in hiPSC liver differentiation are more co-expressed at system level with ERG. |
| TRIM22-(*MHC Class II and I*) | 0.8-0.63 | 15+ | 0.49-0.25 | 15+ | - | 35777501 / 2022 / TRIM22 downregulates MHC-II proteins |
| KSR2-CADPS | 0.81 | 150 | 0.24 | 101 | 1 | 35758323 / 2022 / KSR2 activates CADPS to release insulin in pancreatic islets. |

tissues, is not proportional and seems unrelated, suggesting a possible distinct mechanism. Thus, besides confirming the ASPM and BRCA1 relationship, our tool can also inform exceptions that may help to envision a broader biological picture.

**BRCA1-HMMR.** The top system-level BRCA1 coexpressed gene is HMMR (S2B Fig in S1 File) while in z-score the rank is 71. Their interplay regulates the apicobasal epithelial polarity [41]. Recently, HMMR has been proposed as a risk modifier gene for breast cancer BRCA1 carriers [42]. These shreds of evidence support a coordinated BRCA1-HMMR function validating their coexpression. Interestingly, coexpression also seems lacking in the thyroid gland.

**TP53-CLIC1.** The 10th most system-level coexpressed gene for TP53 is CLIC1 (S2C Fig in S1 File). Given that CLIC1 was previously related to hepatocellular carcinoma, Jiang *et al.* showed that TP53 regulates CLIC1, modulating the MYC pathway [43]. In tissue-level z-score, the CLIC1 is ranked at 165[th], thus difficult to spot.

**DDX58-STAT3.** By overexpression, silencing, and immuno-precipitation of DDX58, it was demonstrated that DDX58 activates STAT3 in a colon cancer cell line [44]. We noted high DDX58-STAT correlation (r = 0.65) ranking 45th at the system-level (S2D Fig in S1 File) while at tissue z-score the rank is higher than 500.

**TWIST1-PDGFRB.** By overexpressing TWIST1, an epithelial to a mesenchymal transition transcription factor, activation of the PDGFRB promoter was observed, which leads to increased proliferation and migration in a breast cancer cell line [45]. In our estimations, the correlation is r = 0.64 ranking 37th at the system-level and 47th at the tissue-level (S2E Fig in S1 File).

**ERG-(Top 1000).** A proteome analysis during the differentiation of induced pluripotent stem cells into liver sinusoidal endothelial cells shows a top list of 30 proteins overexpressed, including ERG [46]. ERG is a transcription factor that in CoGTEx is coexpressed with 22 of these 30 proteins at the system level ($p < 10^{-28}$, hypergeometric test), suggesting that ERG could be a master regulator during the differentiation process. Four examples of ERG are shown in S2F Fig in S1 File.

**TRIM22-(MHC II).** By knocking and overexpressing TRIM22, authors concluded that TRIM22 negatively regulates protein expression of the major histocompatibility complex class II (MHC II) [47]. We noted high positive system-level coexpression with 12 HLA genes of the MHC II (HLA-D) among the top 1000 genes (in Pearson TPM). But also observed 7 HLA genes of the MHC class I (HLA-E, -C, -F, -A, -B, -H, -F-AS). Four examples are shown in S2G Fig in S1 File. In their work, Inoue *et al.* proposed a negative posttranscriptional regulation over MHC II that does not contradict the positive coexpression observed in our estimations at transcription. Positive transcriptional coexpression suggests that proteins involved should be present coordinately to respond to stimuli.

**KSR2-CADPS.** The authors found that acetylcholine-dependent insulin secretion in pancreatic islets depends on CADPS activation by KSR2 [48]. Although the ranking of the estimated system level coexpression is 140, the correlation is high (r = 0.81, S2H Fig in S1 File) and, therefore, important.

## Estimated system-level coexpressions can inform biology by enrichment

Other common use of coexpressed genes is to inform gene functions, biological processes, or pathways in which a gene may be involved. To illustrate this, we performed a gene set enrichment analysis using EnrichR [49] from the top 100 coexpressed genes of BRCA1. Particularly, we used the gene ontology (GO) terms from the top 10 Biological Processes (BP) derived from the top 100 positive coexpressed genes according to the tissue-level estimation compared to the top 10 BP from those coexpressed genes estimated by system-level. Note that a certain

**Table 2. Top 10 GO biological processes for BRCA1 coexpressed genes from an EnrichR analysis from top 100 genes coexpressed at the tissue-level or at the system-level.**

| Term | Rank Tissue-Level | Rank System-Level |
|---|---|---|
| From Tissue-Level coexpression: | | |
| Mitotic Sister Chromatid Segregation | 1 | 1 |
| DNA Metabolic Process | 2 | 19 |
| DNA-templated DNA Replication | 3 | 57 |
| Positive Regulation Of Chromosome Separation | 4 | 2 |
| Regulation Of Chromosome Segregation | 5 | 3 |
| Positive Regulation Of Cell Cycle Process | 6 | 13 |
| Double-Strand Break Repair Via Homologous Recombination | 7 | 34 |
| Double-Strand Break Repair Via Break-Induced Replication | 8 | 76 |
| DNA Synthesis Involved In DNA Repair | 9 | 112 |
| DNA Repair | 10 | 33 |
| From System-Level coexpression: | | |
| Mitotic Sister Chromatid Segregation | 1 | 1 |
| Positive Regulation Of Chromosome Separation | 4 | 2 |
| Regulation Of Chromosome Segregation | 5 | 3 |
| Microtubule Cytoskeleton Organization Involved In Mitosis | 34 | 4 |
| Mitotic Spindle Organization | 26 | 5 |
| Kinetochore Organization | 35 | 6 |
| Regulation Of Chromosome Separation | 11 | 7 |
| Spindle Assembly Checkpoint Signaling | 29 | 8 |
| Mitotic Spindle Assembly Checkpoint Signaling | 30 | 9 |
| Mitotic Spindle Checkpoint Signaling | 31 | 10 |

degree of overlap in the results is expected since 24 genes are present in both lists of top 100 genes, perhaps, at different level of importance. Table 2 shows the top 10 BP resulted in each analysis. Interestingly, using the top genes from the tissue-level estimation, the top BP tended to be related to the important role of BRCA1 in DNA repair of double strand breaks while the top system-level coexpressed genes highlight the role of BRCA1 in spindle-kinetochore-micro-tubule structures [50, 51]. Although many of these BP can be obtained from either, tissue- or system-level rank of genes, this analysis validates the functional information and highlight different aspects of known biology, which can be used for discovery too.

To provide a more comprehensive comparison, we performed the above analysis for C2orf68, MT1M, and NR3C1 genes that were also used for analysis and benchmarking in HGCA 2.0 [29]. In this comparison, we also added the top 100 genes provided by GeneFriends, CoxpresDB, and HGCA 2.0. We merged the top 10 GO BP terms per tool filtering the terms that include at least 3 genes. The results are shown in Fig 7. For C2orf68, our gene rank also associated top terms found by the other three tools related with Skin, Keratinocyte, and Epidermis. There were no exclusive terms found by system-level coexpression, thus, the top biological terms are consistent with the other tools. For MT1M, the EnrichR analysis of positive system-level coexpression rank also associated the top terms found by the other tools related to Zinc, Copper, and other metal ions. There were two exclusive top terms derived from our estimations, but these were observed at the tissue-level coexpression (CoGTEx Z in Fig 7), which were related to Lens Fiber differentiation. Metallothioneins, as MT1M, have been suggested to play a role in the eye [52, 53], perhaps as antioxidant functions [54]. So, the rank from system-level coexpression does not generate GO BP terms (among the top 10) that were not supported by other tools, while the tissue-level provided a sensible prediction. For NRC31, all tools

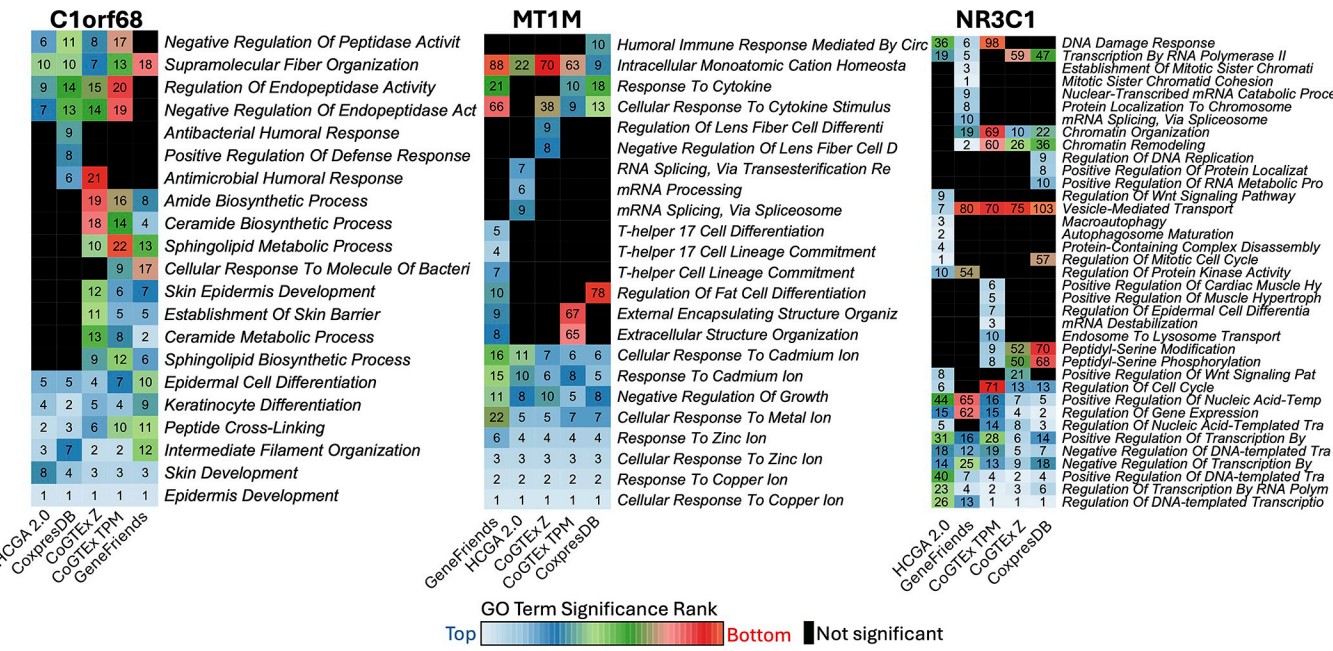

**Fig 7. Comparison of gene set enrichment analyses for three selected genes and four coexpression tools.** Each heatmap shows the rank of significant gene ontology terms for biological processes resulted by a EnrichR analysis from the top 100 coexpressed genes reported by the shown coexpression tools. Colors refer to the position of the rank for the corresponding biological term. Black refers to not significant GO terms or that did not meet the criteria.

yielded exclusive GO BP terms. HCGA 2.0 showed exclusive terms for Autophagosome and Wnt Signalling, GeneFriends for Mitotic Sister Chromatid and mRNA Splicing, CoxpresDB for DNA Replication, RNA Metabolism, and Protein Localization, and system-level coexpression (CoGTEx TPM in Fig 7) showed exclusive terms for Cardiac and Muscle Hypertrophy. A null mutant in zebrafish for the glucocorticoid receptor *nrc31* revealed heart abnormalities [55], suggesting that the association generated by the system-level coexpression could be useful. In summary, this analysis reveals that top system-level coexpressed genes provides sensible and useful gene set enrichment analyses.

## Coexpression is associated to DNA-binding regulation

One of the possible mechanisms for coexpression is the regulation by transcription factors (TF). To explore this possibility, we investigated three TF where the peak results of the chromatin immuno-precipitation sequencing assays (ChIP-Seq) and their corresponding annotated gene are available in GEO. Specifically, we chose BRCA2, ARID4B, and FOXA1. For BRCA2 we used GSE133450 (peaks in file GSE133450_BRCA2.anno.xlsx). For ARID4B, we used GSE148272, (peaks in file GSE148272_K562_ChIP-seq.xlsx), while for FOXA1 we used GSE101407 (the corresponding list of binding peaks in file GSE101407_annotated_1_4_6mer-gehomer_macs _peaks.xlsx). From these, there were 23317, 9860, and 11733 peaks for BRCA2, ARID4B, and FOXA1 respectively. These peaks correspond to 15632, 9860, and 6521 unique genes respectively (in the case of ARID4B, the data was already filtered by duplicated genes). Because the number of peaks in genes are huge, a hypergeometric test of top-coexpressed genes does not represent a full picture of the associations. Therefore, we drawn the ranks of co-expressed genes for Z-Score and TPM scales relative to annotations of peaks compared to the overall ratio of total genes overlapping in both databases (peaks and CoGTEx). As can be seen in the Fig 8, the peaks that suggest a physical interaction is associated to coexpression but

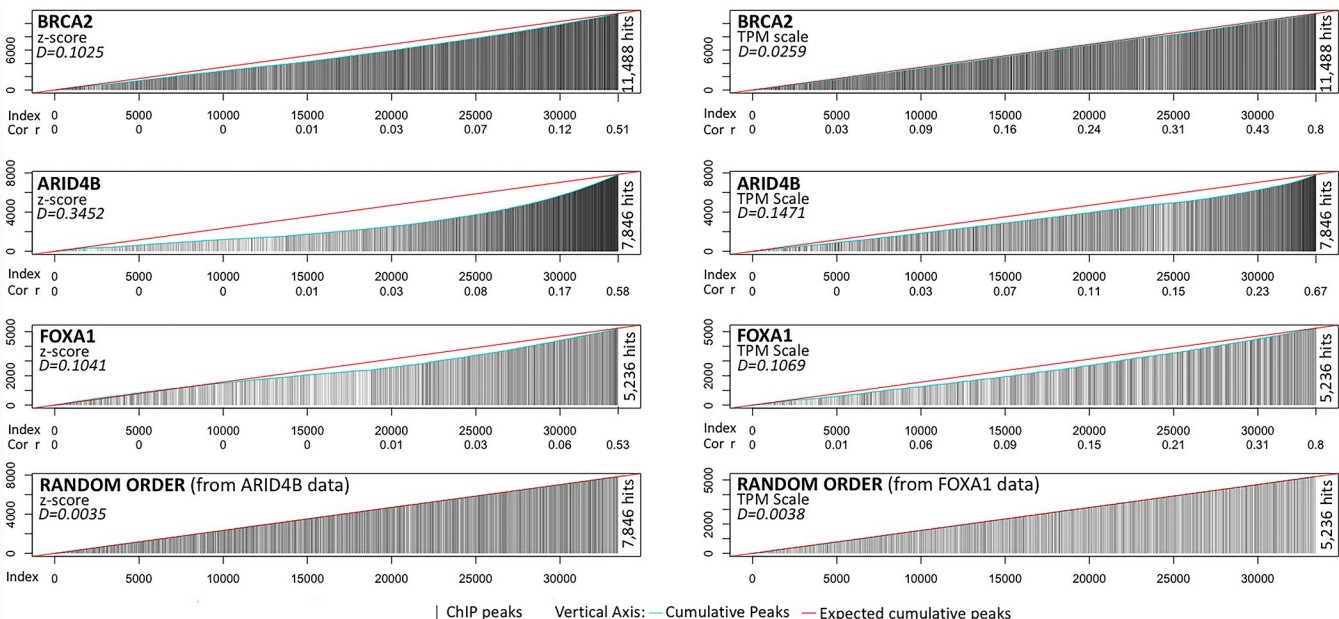

**Fig 8. Comparison of coexpression and ChIP-peaks for selected transcriptional regulators.** Each panel shows the ChIP-peaks for a regulator, in z-score at left, or TPM at right. Vertical black lines represent ChIP-peaks for the corresponding coexpressed gene. The height of each line corresponds to the cumulative number of genes showing a peak. The red line corresponds to the expected ratio, which is estimated by the total genes showing a peak divided by the 33,445 genes. Thus, for random order the height of black lines should match the red line as shown in the bottom panels for two random orders of genes (for ARID4B and FOXA1 data). Deviations from the red line should correspond to higher association between peaks and the order of co-expressed genes. To quantify the deviation, a "D" score was estimated by the sum of the absolute difference between the observed counts (cyan) and the expected counts (red) divided by the total area of the triangle. This D-score represent the fraction of the area that is deviated from the expected. The D-score of ARID4B in z-score is 0.3452 which is almost 100 times higher than random D = 0.0035. The lowest D-score correspond to 0.0259 of BRCA2 in TPM scale, which is 5.5 times higher than random (mean of 100 random order peaks in BRCA2 is D = 0.0047). Note that the three regulators used show deviations from randomness, although at different degrees.

it is highly unspecific, and the magnitude depends on the transcription factor. That is, even that there is some tendency for coexpressed genes to show peaks, it is not definitive. The deviations from randomness (red-line) seems to be TF-specific but also biased by the cell-type used to estimate the ChIP assay. In conclusion, these data cannot be clearly used as ground-truth, mainly, because the number of peaks is huge and unspecific.

## CoGTEx database and web accessibility

The CoGTEx database processed 144 estimates of 559,267,290 coexpression associations (coexpression matrix, CoM) between pairs of 33,445 genes in the GTEx project, including 3 metrics (*Pearson*, *Spearman*, and *G*) and system- and tissue-level modalities. To provide a safe and confident estimation, the minimum estimate observed across these 20 runs was also included and used as the default for further analyses (see Methods). The minimum corresponds to a pessimistic view in which we believe it adds reliability. For the system-level modality, we used quantile-normalized Transcripts Per Million (TPM). For the tissue level modality, we used z-scores per tissue cluster. In summary, we used the minimum of 20 estimations to collapse the 120 CoM sub-samples into 6 CoM representing 3 metrics (*Pearson*, *Spearman*, *G*) in 2 modalities (TPM and z-score). In addition, the database also incorporates one *Pearson* result from samples stratified by sex (2 levels), age (3 levels), and ischemia levels (2 levels), which may be useful to observe association for specific subpopulations (e.g., only females).

The Web application and database are available at http://bioinformatics.mx/cogtex. It follows a simple structure with a home web page to look for genes of interest and a children's

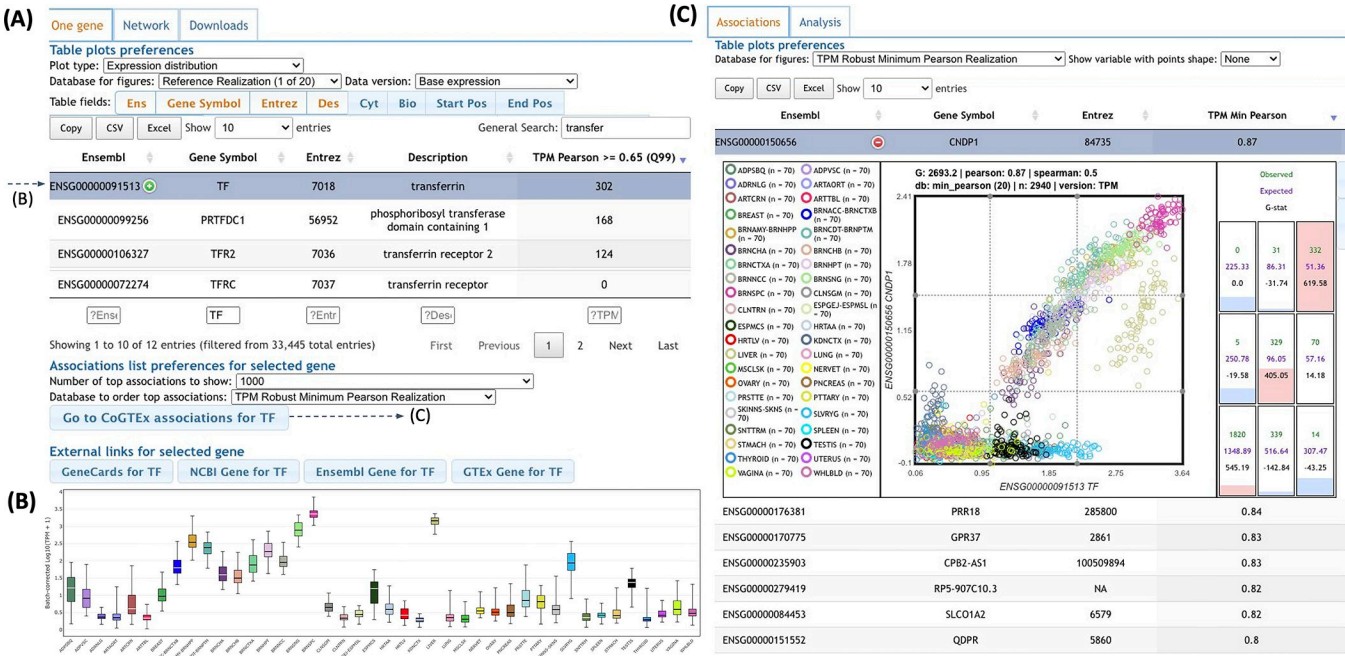

**Fig 9. CoGTEx web database and interface.** Panel A shows the home page where a gene is first selected. A network tab can be used to extract sub-module estimations. Panel B shows an overall expression comparison among tissue clusters. Panel C shows the top/all coexpressions of the selected gene (TF, transferrin gene in the figure). A coexpression figure can be drawn in the interface and contains clickable tissue clusters to remove (right-click) or include (left-click) tissues. The web address is http://bioinformatics.mx/cogtex.

gene list containing the coexpressed genes (Fig 9). The home page is based on a searchable table with the gene symbol, Entrez, and Ensembl gene identifiers, as well as additional gene information (Fig 9A). The expression levels of selected genes across the different tissue clusters of our pipeline can be quickly checked in a plot within the table (Fig 9B). External links to resources elaborating on the gene's info are facilitated upon gene selection. Additionally, the home page also provides tabs to bulk downloads of the data and a network-building tool based on a user-provided set of gene identifiers. The network of associations between the provided genes is displayed in a heatmap format constructed with the Clustergrammer tool [56]. The coexpressed gene list web page is searchable with associations for all metrics. Associations based on covariate levels are also included in this table (see Methods). The listed tables can be easily copied or exported, which facilitate further user analyses such as gene enrichment or network reconstructions. The visual inspection is essential for us to support the estimations. An interactive scatter plot for each association may be easily requested from within the table (Fig 9C). Furthermore, scatter plots for any two genes can be requested for the data giving the final association appearing in the table or for any of the subsampled datasets we used during calculation (see Methods), as well as the source expression data by covariate levels. Scatter plots may be saved to the bottom of the page for quick reference within a browsing session and compared side-by-side on a different webpage. We added a tool to analyze the ranks of computed metrics to identify differences between system- and tissue-level coexpression described in this manuscript (option in Analyze tab within the gene page).

## Discussion

The estimation of coexpression is crucial because it may suggest cooperation or regulation. The standard view of coexpression involves the removal of mean expression levels, potentially

leading to information loss. At least for cooperation, mean transcript levels are quite important. Here, we expand the coexpression concept to estimate coexpression at the system level, where the mean expression level is an essential component. Of course, as in tissue-level coexpression, variation also plays a fundamental role. We have shown that coexpressed genes highly depend on how the coexpression is calculated. Our results show that only 10% to 15% of genes are shared between top system and tissue level coexpression. We also showed that system-level coexpression carries meaningful biological information supported by pathways, tissue networks, gene set enrichment analyses, and literature. Thus, the revision of system-level coexpression opens new opportunities for discoveries.

Contrary to well-studied z-scored coexpression (at the tissue level), our estimation is the first systematic attempt (to our knowledge) to calculate coexpression considering the mean expression level for an organism-wide database such as GTEx. Although we considered a robust estimation using the minimum and a stringent procedure, at a system level, the estimation seems problematic for many genes derived from considering the mean level. For instance, many tissue-specific genes are expressed in one or a few tissues, which generates, wrongly, more coexpressed genes. In this context, we noted many genes highly specific or higher expressed in the testis. Similarly, genes expressed in most tissues except by one or a few, such as BRCC3 or mitochondrial genes where only a few tissues express lower expression levels (such as MT-CO3), also show an excess of wrongly coexpressed genes at the system level. Without a more adequate estimation of coexpression at the system level, which should sort these issues, our estimation is a novel resource for about ⅔ of genes that does not show extreme tissue-specific expression. So, the first revision to use our system-level coexpression is whether the starting gene is not biased by an extreme tissue-expression level. For this, we added a handy box-plot estimation in the web tool and a column estimating the number of tissues in which the gene seems expressed.

While this manuscript was in preparation, the study by Johnson *et al.* was published [57]. They concluded that between-sample normalization has the highest impact on building coexpression networks. Specifically, they showed that quantile normalization might have detrimental effects, perhaps due to extreme value adjustments. Nevertheless, Johnson *et al.* tested tissue-specific networks, and we used a highly robust methods to estimate correlation that considers a subset of samples that may alleviate the extreme-value issue. For example, by looking at system-level coexpressions in Fig 5, it seems rather unlikely that the coexpression estimation will be different when some of the extreme values may be altered by using a different normalization scheme. Thus, it may be possible that the effect observed by Johnson *et al.* does not apply in general in our system-level setting. Moreover, we compared raw data, upper quartile normalization per tissue (as proposed by [58]), quantile-normalization per tissue, and our data after processing (see S1 File) showing that our processing shows comparable results to those normalizations performed at the tissue level. Still, we plan to consider in future versions the Johnson *et al.* recommendations and those from other authors [58–60] focusing on differences compared to the calculations presented here. Based on our robust estimation and that correlation does not seem to be altered by extreme values, we anticipate that differences with current estimations will be minor in the system-level calculations.

As noted by a reviewer, a necessary condition for coexpression at the system level seems to be differential expression among tissues. This interesting view can be used to model system-level gene expression as linear combinations of tissues and individual components. However, differential expression is defined at the single gene level while coexpression is defined at gene pairs, only gene pairs whose differential expression profiles are similar would become coexpressed at the system level. Thus, differential expression and system-level coexpression are two distinct concepts.

Our criteria to filter genes included around 5,300 genes cataloged as pseudogenes (from NCBI gene description). Many of them show high coexpression to protein-coding genes suggesting a well-controlled expression in pseudogenes. For example, the WHAMM pseudogene 1 (ENSG00000223509) is highly coexpressed at the system level with SLC30A7 (Zinc transporter), SCAF11 (Splisosome factor), TBC1B2D (GTPase), and CD40 (TNF receptor), and also negatively correlated with ELOB (Elongation factor), EVIL5L, and PTPA (Protein phosphatase). Thus, revision of particular cases may provide biological insights.

Our coexpression estimation could be affected by other factors. For example, it is known that ancestry has an impact on gene expression [61]. In this context, the ancestry effect seems to be low (median of 0.35 on the logarithmic scale) and it involves a small fraction of genes (~1%). We use the same individuals in both genes in our coexpression estimation, and we used a robust estimation procedure that does not depend on particular subjects. These factors suggest that most coexpression estimations should not be affected by ancestry, nevertheless, it remains unknown the extent that it does affect system-level coexpression estimations or whether particular populations show differences in system-level coexpression networks. Similarly, it is also known that particular genetic variants affect gene expression [24]. Although these expression quantitative trait loci (eQTL) may have influenced specific coexpressions, we believe the effect in the overall coexpression estimation would be low. This is supported by the fact that eQTLs affect a small fraction of the individuals (otherwise, it is a common variant absorbed in the overall expression) and by the small number of eQTL per gene (~2 eQTL/gene) [24]. In addition, single-cell RNA-Seq analyses have shown that tissues are composed of a mixture of cells in different proportions [62]. Thus, our estimations derived from GTEx corresponding to bulk tissue RNA data represent an overall approximation. Nevertheless, in the context of single-cell RNA-Seq, our data can be used, for example, to answer whether the overall coexpression agrees with single-cell clusters; in any case, it could reveal interesting intra-tissue mechanisms. Also, there is a bias in the number of samples per tissue in the GTEx data. In the 20 rounds of estimation, we randomly sampled 70 individuals. Thus, if the coexpression depends heavily on a small-sampled tissue, it may be biased by the use of similar sets of individuals.

We showed that transcriptional regulation is partially associated to system- and tissue-level coexpression. This result shows that some coexpressed genes are due to direct genetic regulation (e.g. TF→Gene). However, TF-binding or regulator bound to TFs seems highly unspecific, which could be due to the cell-type context or stimuli and the result of indirect or convergent genetic regulations. The indirect regulation occurring when a regulator appears coexpressed with a *Gene* through a second regulator (e.g. TF1→TF2→Gene) while the convergent resulting when *Gene1* is coexpressed with *Gene2* due to similar regulation (e.g. TF→Gene1 and TF→Gene2). In any case, system- and tissue-level coexpressions could provide clues or hypotheses of such regulations.

For us, viewing the coexpression among genes is very important because it supports the coexpression estimation and the tissues where it is estimated. Thus, our tool focus on showing these associations, which is also useful to export figures and promote research. Other tools, such as GeneFriends [28], do not allow to show coexpression. We plan to update our tool regularly to help explore system and tissue level coexpression easier and more meaningful.

What are the mechanisms leading to system-level coexpression? As in tissue-level coexpression, the expectative is that there is a genetic control behind. We have shown examples where the mean level, and therefore system-level coexpression, is important (Fig 5 and S1 File). By now, we believe our database will be useful to provide insights and hypotheses, which may lead to help uncover novel biological mechanisms.

We cannot declare that system-level coexpression outperform other coexpression calculations, in particular those of z-score estimations as CoXPRESdb. Although it can be tested whether different estimations of coexpression are more precise than others, as we did with pathway databases, all these comparisons are based on a known reference. The reference is the cummulative knowledge of interactions or regulations and, in pathway cases, even knowledge that is selected for particular purposes. Nevertheless, this reference knowledge is far from perfect and is highly biased to how the knowledge is being accumulated. For example, many experiments are based on experimental assays within a few cell-lines or tissues (similar to tissue-based estimations or z-scores) instead of what happen in an organism as a whole. Thus, we do not have a comprehensive ground truth of gene coexpression to accurately evaluate tools. Indeed, our results confirm that z-scores per tissue seems more accurate than TPM estimations. So, basically CoXPRESdb (or per-tissue z-score estimations) should outperform system-level estimation in terms of accuracy compared to current knowledge. Nevertheless, this does not undermine our proposal. With the recent acquisitions of massive experiments and efforts (such as those from GTEx), novel approaches are needed to view the data in different ways. We show that our system-level estimation provides a different view, that of observing an organism as a whole and the aspects of gene expression along all tissues in the organism as opposite to observing a particular tissue or cell-line, or relative expression across tissues. We show that our estimations makes sense and we hope that our view and accompaining tool will help to discover more biological knowledge. In summary, the content and estimations of coexpression are different between CoGTEX and common tools for tissue-relative coexpression estimation from GTEx (such as CoXPRESdb [20, 38], HGCA [29], GeneFriends [28]) and other tools such as StringDB [63]. Our focus is to show that the estimation provided by CoGTEX are different and useful.

By the examples shown and because our proposal provides novel, highly related functional gene partners that can be useful in discovering normal and abnormal molecular mechanisms, we conclude that our system-level estimation can generate reasonable and verifiable biological hypotheses thus it is valuable and could positively impact biological health and disease research.

## Methods

### Data collection

Version 8 of gene-level TPM expression data and free-access metadata was downloaded from the GTEx website on May 23, 2021 (https://gtexportal.org/home/datasets files "GTEx_Analysis_2017-06-05_v8_RNASeQCv1.1.9_gene_tpm.gct.gz", "GTEx_Analysis_v8_Annotations_SampleAttributesDS.txt" and "GTEx_Analysis_v8_Annotations_SubjectPhenotypesDS.txt"). The collected data originally had 17,382 RNA-Seq samples from 54 tissues (2 represent duplicates of 2 other tissues with different preservation methods and ischemic time, https://gtexportal.org/home/faq). After data collection, we excluded tissues with sample counts less than 50 (21 bladder, 9 ectocervix, 10 endocervix, 9 fallopian tube, 4 kidney medulla). EBV-transformed lymphocytes (174) and cultured fibroblasts (504) were also excluded [24], leaving 16,651 samples from 47 tissues.

### Gene selection

Low-expressed genes were filtered out as they can represent noisy measurements [64]. For this, we used an assessment employed by GTEx [24] that tests if each gene is expressed at a value of 0.1 TPM or more in at least 20% of samples (3,331 samples). 24,720 genes failed this test and were initially considered low-expressed (31,480 passed). However, we anticipated that some of these genes failed the test due to a degree of tissue-specificity as the procedure

naturally retains only genes expressed across several tissues. Briefly, for all genes filtered out in the previous test, we calculated the mean expression and the percentage of samples expressed at 1 TPM or more per tissue. We considered the maximum tissue mean expression value greater than 5 TPM combined with a maximum percentage of expressed samples greater than 66% per tissue as evidence that a gene should be retained. This adds certainty that a gene is clearly not low-expressed in at least one tissue. With this procedure, we rescued 1,955 genes from the 24,720 that had been filtered out initially, leaving a final count of 33,445 genes for downstream analyses.

## Data normalization and correction

For normalization, we first variance-stabilized the data by adding a pseudo count of 1 to all TPM values and computing the logarithm base 10 of these quantities (as in the GTEx portal). We followed by applying quantile normalization [65] (*preprocessCore* R package). Although many methods exist to adjust gene expression for explicit and hidden covariates, using classical batch correction techniques performs well for downstream coexpression analysis [66]. Therefore, we used ComBat, a bayesian framework for batch correction of the data after normalization [67]. Since there are two batch covariates in the GTEx metadata, ComBat was run in series for the extraction batch first and then for the sequencing batch. Samples with incomplete batch information or that belonged to a batch with less than two samples total were assigned placeholder batches. Default parameters of the R *sva* package ComBat implementation were used, and the tissue, sex, and age-matched with each sample were indicated as variables to preserve during the batch correction.

## Estimation of purified GTEx tissue clusters

To handle possible artifacts in RNA-seq gene expression due to sample acquisition, sample preparation, sequencing, and unknown effects, we performed a two-stage strategy to identify possible outlying samples, as shown in S3 Fig in S1 File. In the first stage, we used principal component (PC) analysis summarizing the inter-tissue gene expression variability retaining 159 statistically significant PC explaining 79% of the variance from 16,651 samples. Then, we used the estimated PC in a Louvain algorithm based on k-nearest-neighbors (kNN) clustering samples by similitude to identify samples that could be mixed among several tissues (n = 1066) or associated with a divergent tissue (n = 593). Overall, this first-level procedure removed 1659 samples (marked in pink or orange respectively in S3 Fig in S1 File). The second stage consists in purifying clusters by another kNN procedure based on the t-SNE coordinates (S4 Fig in S1 File). First, we noted five tissue pairs whose samples were clearly mixed, so these tissues were merged to generate tissue clusters and named from the original tissues. Then, we removed those samples appearing far from their t-SNE cluster. This procedure removed 135 additional samples (marked in yellow in S3 Fig in S1 File). The final dataset comprised 14857 samples among 42 tissue clusters. The filtered set of samples represents a clear sample tissue-clustering according to tSNE (S4 Fig in S1 File). The details of these two-stages procedures are shown in the following three paragraphs.

## Principal component (PC) analysis (PCA) of GTEx samples

After standardization (m = 0, sd = 1), we applied PCA [68] to the full 16,651 samples by 33,445 genes (*gmodels* R package). To determine how many PC to retain, we used a 100 permutations test [69] sampling the explained variance by the first PC as implemented in the *Rspectra* R package. This procedure resulted in 159 PCs retained, accumulating 79.2% of the variance in the original data.

## Clustering by k-nearest-neighbours

The Louvain algorithm (*igraph* R package) was applied for clustering. The input is a graph where each sample is connected to its k-nearest neighbors. We weighted the edges using the Jaccard index between the sets of neighbors of any pair of samples. We constructed 100 graphs varying the number of neighbors k from 1 to 100 to explore different possible partitions of the data. Graphs were based on the Euclidean distance in PC space. To obtain the best value of k for each tissue, we used P = (I/T) * (I/C), where I is the intersection of samples in tissue T and cluster C. Then, formed clusters were used to identify and remove samples mixed with other tissues or not assigned to a cluster.

## t-SNE and cluster outlier removal

To visualize samples and their tissue or cluster memberships, we computed a 2-dimensional t-SNE using the Barnes-Hut [70] implementation in the *Rtsne* R package. We applied the algorithm to all samples before the filtering process in PC space. The perplexity of 30, the learning rate of 200, theta of 0, the maximum number of iterations of 1000, no PCA, and no normalization were used as parameters. Other parameters not mentioned were left as defaults. After removing samples intersecting between clusters and samples with a tissue-cluster mismatch, as explained before, we used the t-SNE coordinates of the remaining samples to visualize outliers. We found that applying the following procedure removed most outliers: compute the medoids (removing "boxplot outliers") in any of the two dimensions of the t-SNE, then identify the closest medoid to each data point, and finally predict the cluster membership by a voting *k*-nearest neighbors approach where *k* is determined by the size of the cluster whose medoid is closer to each data point. If the cluster membership differed from the prediction in this procedure, the sample was removed. This process removes 125 samples. Additionally, we manually removed 10 more visual outliers that remained even after the procedure.

## Gene association measures

We used the common *Pearson Correlation Coefficient*, the *Spearman Rank Correlation Coefficient*, and a modified version of the *G-statistic* [31, 32]. *G* is inspired by the Maximal Information Coefficient (MIC) to capture possible non-linear behaviors but with less intensive computation costs at the expense of working with a fixed data discretization [39]. The *G-statistic* is similar to the $\chi^2$ contingency test. So, the expression values of each gene are first independently discretized into three bins of low, medium, and high expression, respectively. This is done via the *K*-means algorithm initialized with three deterministic centroids at the minimum (quantile=0.2), median (quantile=50), and maximum (quantile=99.8) values per gene, removing those below and above 0.2 and 99.8 quantiles. A maximum of 1000 iterations for most genes was enough to achieve convergence using the Hartigan and Wong [71] implementation of *K*-means in the R *stats* package. Genes that did not reach convergence this way did with the Lloyd [72] implementation in the same package after setting the maximum iterations to 10,000. All other settings were left as default. These three discrete levels were used to estimate the *G-statistic* as $G=2 \sum O_{ij} * ln(O_{ij}/E_{ij})$ where $i$ and $j$ are indexes of the three discrete levels, $O$ represents the "observed" counts, and $E$ the expected ones as in a $\chi^2$ contingency test (the product of marginals of the involved categories divided by the sum of all counts). In preliminary runs, we observed very large G values caused by low $O$ and infinitesimal $E$ values; thus, we introduced a small correction [73]. The minimum $E_{ij}$ value was set to 1 or 0.5, depending on whether $O_{ij}$ was larger than four. We show that the distribution of this correction has imperceptible effects in the null distribution [74]. In scatter plots on the CoGTEx web service, we provide the contingency tables for each pair of genes together with post-hoc P-value estimates for each cell computed from standardized residuals [75].

## Database calculation process

To provide a robust estimate of associations between pairs of genes, we employed a calculation strategy based on random sampling and repeated computations. For each gene expression state represented by a cluster in our pipeline, we sample uniformly at random 70 samples without repetition. This number was selected based on ~80% of the size of our smallest cluster. The final coexpression value for any pair of genes is determined by examining all values that involve the pair across the 20 runs of a particular association measure and taking the minimum.

## Standardized database version

Additional to the main CoGTEx database, which is calculated from TPM-scale gene expression values, we have also repeated calculations using standardized data by tissue cluster for comparative purposes with other databases such as COXPRESdb. This is referred as Z-Scores in our web application. For this, TPM gene expression in log10 scale was transformed to z-scores by $z = (x - m) / s$, where $m$ is the mean expression of the gene in a particular tissue, and $s$ is the sample standard deviation of the gene in the tissue, and $x$ represent the gene expression of a sample in the tissue.

## Associations by covariate levels

CoGTEx includes *Pearson* estimates for combined distinct gene expression states in the human body stratified by covariate levels. For SEX, the levels are male and female. For AGE, the levels are less than 50 years old, between 50 and 59 years old, and more than 59 years old. For ischemia time (SMTSISCH), the levels are low ischemia and high ischemia. For this task, k-means centroids were initialized at the minimum and maximum values of the vector of ischemia times. For each covariate level, we assembled a unique expression matrix with the appropriate number of samples per cluster and proceeded to downstream computations. A z-score version of this estimation was similarly computed as described in the standardized database version section.

## Functional enrichment analysis

Canonical KEGG [76] pathways were retrieved using the *gage* R package [34]. A total of 11,053 genes across 176 pathways containing at least 2 genes each were used. For each gene in CoGTEx, we retrieved the list of associations of the gene with all others in the database. After sorting the list from strongest to weakest coexpression, the top and bottom 5% of associations were treated as gene sets for enrichment analysis. Enrichment was estimated using a hypergeometric test to assess if the overlap between the top or bottom gene sets and each of the KEGG pathways was significant. For each gene, we recorded the number of hypergeometric tests yielding a p-value lesser than 0.05. The GIANT [35] "all tissues" and all 144 tissue-specific full networks were obtained from http://giant.princeton.edu/download/ on August 2021. To determine which gene pairs from these networks we would consider as "real" positives and negatives to compare to our coexpression estimations, we use a procedure similar to the benchmark described by Somekh *et al*. [66]. Briefly, the probability of association of all edges in the networks involving pairs of genes in our database was examined. For each network, if an edge represented a probability higher than 0.8 of a gene pair to be associated, it was considered a real positive. For the real negatives, we chose at random from the edges with a probability lesser than 0.005 an equal number of pairs as the positive examples per network. This resulted in balanced positives and negatives for most of the networks. For each combination of network

and coexpression metrics in our database, we computed a ROC curve using 1000 distinct operation points represented by coexpression thresholds.

### Analyses of chromatin immuno-precipitation assays with sequencing

Data from GEO were used to extract peaks representing DNA binding for transcriptional regulators. The series files GSE133450 (peaks in file GSE133450_BRCA2.anno.xlsx), GSE148272, (peaks in file GSE148272_K562_ChIP-seq.xlsx), and GSE101407 (the corresponding list of binding peaks in file GSE101407_annotated_1_4_6mergehomer_macs _peaks.xlsx) were used for BRCA2, ARID4B, and FOXA1 respectively. From these data, there were 23317, 9860, and 11733 peaks for BRCA2, ARID4B, and FOXA1 respectively. These peaks correspond to 15632, 9860, and 6521 unique genes respectively (in the case of ARID4B, the data was already filtered by duplicated genes). Finally, from these annotated genes there were 11488, 7846, and 5236 genes respectively that were also included in our 33,445 genes analyzed for coexpression. For each regulator, the CoGTEx list of all coexpressed genes were extracted from the web tool and sorted by absolute value. Then, the gene symbol was matched to the corresponding peak list and annotated whether the gene was present or not. Cumulative presence counts were then used and showed in figures.

## Supporting information

**S1 File. Includes supplementary methods and 6 supplementary figures.**
(DOCX)

## Author Contributions

**Conceptualization:** Víctor Treviño.

**Data curation:** Miguel-Angel Cortes-Guzman, Víctor Treviño.

**Formal analysis:** Miguel-Angel Cortes-Guzman, Víctor Treviño.

**Funding acquisition:** Víctor Treviño.

**Investigation:** Miguel-Angel Cortes-Guzman, Víctor Treviño.

**Methodology:** Miguel-Angel Cortes-Guzman, Víctor Treviño.

**Project administration:** Víctor Treviño.

**Resources:** Víctor Treviño.

**Software:** Miguel-Angel Cortes-Guzman, Víctor Treviño.

**Supervision:** Víctor Treviño.

**Validation:** Víctor Treviño.

**Visualization:** Miguel-Angel Cortes-Guzman, Víctor Treviño.

**Writing – original draft:** Miguel-Angel Cortes-Guzman, Víctor Treviño.

**Writing – review & editing:** Miguel-Angel Cortes-Guzman, Víctor Treviño.

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
