## [Decision Letter · Decision Letter 0]

16 Jul 2024

PONE-D-24-02309CoGTEx: Unscaled system-level coexpression estimation from GTEx data forecast novel functional gene partnersPLOS ONE

Dear Dr. Trevino,

Thank you for submitting your manuscript to PLOS ONE. After careful consideration, we feel that it has merit but does not fully meet PLOS ONE’s publication criteria as it currently stands. Therefore, we invite you to submit a revised version of the manuscript that addresses the points raised during the review process. You are required to address all the comments by the reviewer, either by performing additional analyses, or by explaining why you believe these are unnecessary or unfeasible. The reviewer's suggestions about the web site should be considered as non-mandatory.

We look forward to receiving your revised manuscript.

Kind regards,

Paolo Provero, Ph.D.

Academic Editor

PLOS ONE

4. We note that Figures 1 and 2 in your submission contain copyrighted images. All PLOS content is published under the Creative Commons Attribution License (CC BY 4.0), which means that the manuscript, images, and Supporting Information files will be freely available online, and any third party is permitted to access, download, copy, distribute, and use these materials in any way, even commercially, with proper attribution. For more information, see our copyright guidelines: http://journals.plos.org/plosone/s/licenses-and-copyright.

1. You may seek permission from the original copyright holder of Figures 1 and 2 to publish the content specifically under the CC BY 4.0 license.

Reviewers' comments:

Reviewer's Responses to Questions

**Comments to the Author**

1. Is the manuscript technically sound, and do the data support the conclusions?

Reviewer #1: Partly

2. Has the statistical analysis been performed appropriately and rigorously? 

Reviewer #1: Yes

3. Have the authors made all data underlying the findings in their manuscript fully available?

Reviewer #1: Yes

4. Is the manuscript presented in an intelligible fashion and written in standard English?

Reviewer #1: Yes

5. Review Comments to the Author

Reviewer #1: The manuscript of Cortes-Guzman et al describes a large-scale gene coexpression analysis based on GTEx v8 gene expression data and the development of an online webtool for the exploration of results. The strong points of this work are the in-depth quality control and preprocessing of the GTEx data, as well as the calculation of global and tissue-level correlations using multiple metrics and association with other traits.

The manuscript is overall well-written with a sound methodology, however I have some comments regarding the authors selection of normalisation method and lack of comparisons with more coexpression webtools, apart from COXPRESdb. Please, find my comments in detail below:

Major Comments

i) The authors performed an elaborate pipeline for both gene and sample filtering. Indeed, quality-controlled samples result in higher quality coexpression networks. However, in this case, it is difficult to gage how much the preprocessing procedure aided towards that goal, as the expression data were initially normalised using Quantile Normalisation. The authors themselves present Johnson et al. [48] conclusions as follows: "between-sample normalization has the highest impact on building coexpression networks. Specifically, they showed that quantile normalization might have detrimental effects, perhaps due to extreme value adjustments.". Additionally, Hicks et al. (https://doi.org/10.1186/s13059-015-0679-0) mention that applying global normalisation methods (such as quantile normalisation) in samples of different tissues has the potential to remove biologically driven variation, leading to increased bias and mean squared error in downstream analyses. Moreover, Vandenbon (https://doi.org/10.1371/journal.pone.0263344) showed that Quantile Normalisation preprocessing resulted in the lowest-quality coexpression networks overall. Thus, the authors should consider another normalisation method, perhaps a per-tissue method, such as smooth quantile normalisation or tissue-aware normalisation (https://doi.org/10.1186/s12859-017-1847-x) also tested on GTEx data, and whether that could improve the correlations and, as such, the performance of their tool.

ii) The authors performed a comparison of their correlations with those of COXPRESdb, another popular coexpression database, checking the overlaps of the top 100 partners for each gene, between COXPRESdb and CoGTEx, and showing only a 15% average gene overlap. Is it possible for the authors to comment whether their method ultimately outperforms the one used by COXPRESdb?

Furthermore, the authors mention several GTEx-based coexpression works in the introduction, claiming that "researchers have no resources to view and analyze GTEx gene coexpression partners easily". However, webtools, such as GeneFriends (mentioned in the manuscript) and HGCA (not mentioned in the manuscript, https://doi.org/10.3390/cells12030388), offer results from large-scale gene coexpression analyses based on the same GTEx data, while also offering visualisation options (Networks or Dendrograms respectively), and seamless enrichment analyses. Since the authors already used COXPRESdb in benchmarking, it would be interesting to see how CoGTEx fares compared to those two other GTEx-based webtools.

iii) For their Enrichment benchmarking, the authors could also consider the three aspects of Gene Ontology, apart from KEGG pathways and GIANT networks. Furthermore, in the results section "Estimated system-level coexpressions can be validated", the authors could enhance their examples by adding external enrichment analysis of multiple coexpressed partners (adding enriched GO terms or Pathways), rather than just focusing on one-pair relationships. As the main aim of coexpression webtools is to replicate known biology, the best way to ascertain the validity of the coexpressed gene partners to a single gene, it to see whether the enriched biological terms of the coexpressed genes are related to the bibliography-supported input gene's biological functions. In any case, we expect the authors to comment on the relevance of the enriched KEGG terms of the top coexpressed genes and the selected gene.

iv) Could the authors elaborate more on how they computed the z-score for the tissue-level modalities in the methods section "Standardized database version"?

v) The authors should clarify in the text when the signed and when the absolute correlations are used. For example, in the method section "Enrichment of biological terms supports coexpression estimations", it is not clear if the top and bottom coexpressed partners are determined using the absolute correlation values or if the anti-correlated genes (negative correlations) are considered as the least coexpressed ones. Same applies to the web platform itself.

Web platform

i) The web interface is simple and fast, however, I was not able to get the Coexpressed genes rank comparison plot to work.

ii) In addition, it would be nice if the authors could add a link or button to automatically submit the list of coexpressed genes to other websites for enrichment analysis, such as StringDB or EnrichR.

iii) Finally, adding a tutorial or FAQ to guide less technologically adept users (that would include Biologists who would be the main users of the webtool) would also be nice.

Minor Comments

i) The authors should update their citations for all webtools and databases mentioned, to match the latest corresponding publications. One such example is COXPRESdb, with the most recent publication being the following: https://doi.org/10.1093/nar/gkac983

ii) The authors should perform a more thorough proof-reading of the manuscript as there are certain typos, e.g. CoGTEx is spelled as CoGTex, CoGTeX and CoGTEX in multiple cases.

6. PLOS authors have the option to publish the peer review history of their article (what does this mean?). If published, this will include your full peer review and any attached files.

Reviewer #1: No

---

## [Author Response · Author response to Decision Letter 0]

21 Jul 2024

We have answered all reviewers' comments. We have submitted new files for manuscript, manuscript tracked, rebuttal letter, response to reviewers, and updated the supporting information.

---

## [Decision Letter · Decision Letter 1]

6 Aug 2024

PONE-D-24-02309R1CoGTEx: Unscaled system-level coexpression estimation from GTEx data forecast novel functional gene partnersPLOS ONE

Dear Dr. Trevino,

Thank you for submitting your manuscript to PLOS ONE. After careful consideration, we feel that it has merit but does not fully meet PLOS ONE’s publication criteria as it currently stands. Therefore, we invite you to submit a revised version of the manuscript that addresses the points raised during the review process.

As you can see, the reviewer believes their comments have been addressed only partially. Please address all the remaining comments, or explain why this is not useful/feasible. Regarding the comments about the accompanying web site, these seem to imply that the tool is not performing as promised, so these should be addressed as well. Please submit your revised manuscript by Sep 20 2024 11:59PM. If you will need more time than this to complete your revisions, please reply to this message or contact the journal office at plosone@plos.org. Please include the following items when submitting your revised manuscript:A rebuttal letter that responds to each point raised by the academic editor and reviewer(s). You should upload this letter as a separate file labeled 'Response to Reviewers'.A marked-up copy of your manuscript that highlights changes made to the original version. You should upload this as a separate file labeled 'Revised Manuscript with Track Changes'.An unmarked version of your revised paper without tracked changes. You should upload this as a separate file labeled 'Manuscript'.

We look forward to receiving your revised manuscript.

Kind regards,

Paolo Provero, Ph.D.

Academic Editor

PLOS ONE

Reviewers' comments:

Reviewer's Responses to Questions

**Comments to the Author**

1. If the authors have adequately addressed your comments raised in a previous round of review and you feel that this manuscript is now acceptable for publication, you may indicate that here to bypass the “Comments to the Author” section, enter your conflict of interest statement in the “Confidential to Editor” section, and submit your "Accept" recommendation.

Reviewer #1: (No Response)

2. Is the manuscript technically sound, and do the data support the conclusions?

Reviewer #1: Partly

3. Has the statistical analysis been performed appropriately and rigorously? 

Reviewer #1: No

4. Have the authors made all data underlying the findings in their manuscript fully available?

Reviewer #1: Yes

5. Is the manuscript presented in an intelligible fashion and written in standard English?

Reviewer #1: Yes

6. Review Comments to the Author

Reviewer #1: I have read the revised version of Cortes-Guzman et al. The authors additionally performed an extensive analysis showing how their normalisation method fares with other normalisation methods shown to perform well for coexpression analysis. In addition, they added a global and local GO term analysis for BRCA1 coexpressed genes of CoGTEx.

However, in their response about the benchmarking with other similar webtools (GeneFriends and HGCA), they only studied one gene pair (TBX19 and POMC). My major comment lies with this and the subsequent comparison with other webtools that I originally asked for.

1) First of all, the authors only mentioned the interaction between TBX19 and POMC in the "response to reviewers" but they did not include this information in the text. The authors did not give any explanation as to why they selected that specific gene pair (TBX19 and POMC). Indeed, TBX19 is a tissue-specific regulator for POMC (PMID: 11447259). The coexpression of TBX19 and POMC is loosely supported in StringDB (https://string-db.org/interaction/9606.ENSP00000356795/9606.ENSP00000384092?c1=65ffa3&c2=ff0000), as putative homologs were coexpressed in other organisms. The authors reported that TBX19 and POMC had a 0.42 Pearson correlation in CoGTEx and ~0.88 in GeneFriends, but they failed to realize that HGCA did not discover this gene correlation as significant, as they expected the actual calculations of coexpression between genes.

Ultimately, the things that matter to the end users (Molecular Biologists) when they use a coexpression tool to identify novel functional partners to a gene of interest, are the top ranked genes of a coexpressed gene list and the correlation cutoff, not the data and procedures that resulted in that list.

In CoGTEx, when the coexpressed genes for TBX19 are downloaded, POMC is ranked in 1020th position, while in the corresponding list for POMC, TBX19 is ranked as 185th. In their "response to reviewers", the authors do not study the top coexpressed partners of each of the two genes and whether these are bibliographically-supported functional partners, thus not stating if they consider that the 0.42 correlation of CoGTEx to TBX19 and POMC is something that should be expected or not.

To infer whether a webtool has the potential to identify new functional relationships, we can check the relationships between the gene of interest and its coexpressed genes one-by-one. The authors have already performed this, through a PubMed literature search approach. However, this method is not easily verifiable and it cannot be massively performed for the benchmarking of different webtools.

Instead, to facilitate the process of confirming whether the tool can replicate known biology, we can check if the enriched biological terms in the list of top coexpressed genes are related to the gene of interest’s functions. This is easy to perform for multiple webtools. The authors have already used that approach on the top 100 local and global coexpressed partners of BRCA1, only for their own tool. In my previous comments, when referring to benchmarking, I was expecting the authors to do the same thing, for the other similar webtools, and not just mention one single gene pair (TBX19 and POMC).

Taking all this into account, the authors should perform this benchmarking with CoGTEx and the other webtools (CoXPRESdb, GeneFriends and HGCA) by taking the top 100 or so coexpressed partners in each webtool, for 3-4 cases of genes with well-documented roles in bibliography. The authors can use EnrichR or any other enrichment webtool (StringDB, gProfiler, WebGestalt). The authors can refer to the paper of HGCA (PMID: 36766730), where that kind of benchmarking was performed (See Section 2.6. STRING Analysis and Section 3.2. Coexpression Tool Benchmarking).

2) In their “response to reviewers”, the authors added a rationale for the difference in results with CoXPRESdb. They should add something like that in the discussion section of the main text, also including the other webtools they will use in the benchmark.

Minor Comment:

1) It is still not possible to run the "coexpressed gene rank comparison" plot in CoGTEx. It seems to load indefinitely when I choose "coexpressed gene rank comparison" option in the plot type and click the green cross icon next to the ENSG of a gene. I did not ask for the authors to amend the text, but to solve this issue in their website. Also, I am unable to find within the website what the authors mention as "option in Analyze tab within the gene page", unless they refer to the green cross icon next to each gene ENSG code.

7. PLOS authors have the option to publish the peer review history of their article (what does this mean?). If published, this will include your full peer review and any attached files.

Reviewer #1: No

---

## [Author Response · Author response to Decision Letter 1]

14 Aug 2024

Please refer to the submitted PDF document "Point-by-point answers CoGTEX PLOS-One-rev4.pdf" where we successfully answer all reviewers concerns.

---

## [Decision Letter · Decision Letter 2]

22 Aug 2024

CoGTEx: Unscaled system-level coexpression estimation from GTEx data forecast novel functional gene partners

PONE-D-24-02309R2

Dear Dr. Trevino,

We’re pleased to inform you that your manuscript has been judged scientifically suitable for publication and will be formally accepted for publication once it meets all outstanding technical requirements.

Kind regards,

Paolo Provero, Ph.D.

Academic Editor

PLOS ONE

Additional Editor Comments (optional):

Reviewers' comments:

Reviewer's Responses to Questions

**Comments to the Author**

1. If the authors have adequately addressed your comments raised in a previous round of review and you feel that this manuscript is now acceptable for publication, you may indicate that here to bypass the “Comments to the Author” section, enter your conflict of interest statement in the “Confidential to Editor” section, and submit your "Accept" recommendation.

Reviewer #1: All comments have been addressed

2. Is the manuscript technically sound, and do the data support the conclusions?

Reviewer #1: Yes

3. Has the statistical analysis been performed appropriately and rigorously? 

Reviewer #1: Yes

4. Have the authors made all data underlying the findings in their manuscript fully available?

Reviewer #1: Yes

5. Is the manuscript presented in an intelligible fashion and written in standard English?

Reviewer #1: Yes

6. Review Comments to the Author

Reviewer #1: (No Response)

7. PLOS authors have the option to publish the peer review history of their article (what does this mean?). If published, this will include your full peer review and any attached files.

Reviewer #1: No

---

## [Editor Report · Acceptance letter]

26 Sep 2024

PONE-D-24-02309R2 

PLOS ONE

Dear Dr. Trevino, 

I'm pleased to inform you that your manuscript has been deemed suitable for publication in PLOS ONE. Congratulations! Your manuscript is now being handed over to our production team.

Kind regards, 

on behalf of

Dr. Paolo Provero 

Academic Editor

PLOS ONE